# Deep Tower Networks for Efficient Temperature Forecasting from Multiple Data Sources

**DOI:** 10.3390/s22072802

**Published:** 2022-04-06

**Authors:** Siri S. Eide, Michael A. Riegler, Hugo L. Hammer, John Bjørnar Bremnes

**Affiliations:** 1Norwegian Meteorological Institute, 0313 Oslo, Norway; johnbb@met.no; 2OsloMet, 0167 Oslo, Norway; hugoh@oslomet.no; 3SimulaMet, 0167 Oslo, Norway; michael@simula.no; 4Department of Computer Science, University of Tromsø, 9019 Tromsø, Norway

**Keywords:** tower network, temperature forecasting, video prediction, deep learning

## Abstract

Many data related problems involve handling multiple data streams of different types at the same time. These problems are both complex and challenging, and researchers often end up using only one modality or combining them via a late fusion based approach. To tackle this challenge, we develop and investigate the usefulness of a novel deep learning method called tower networks. This method is able to learn from multiple input data sources at once. We apply the tower network to the problem of short-term temperature forecasting. First, we compare our method to a number of meteorological baselines and simple statistical approaches. Further, we compare the tower network with two core network architectures that are often used, namely the convolutional neural network (CNN) and convolutional long short-term memory (convLSTM). The methods are compared for the task of weather forecasting performance, and the deep learning methods are also compared in terms of memory usage and training time. The tower network performs well in comparison both with the meteorological baselines, and with the other core architectures. Compared with the state-of-the-art operational Norwegian weather forecasting service, yr.no, the tower network has an overall 11% smaller root mean squared forecasting error. For the core architectures, the tower network documents competitive performance and proofs to be more robust compared to CNN and convLSTM models.

## 1. Introduction

Weather prediction can be seen as a complex problem which often requires methods that are able to incorporate different sources of data at the same time. This can include, for example, satellite images and physical properties of the atmosphere, e.g., temperature, moisture, pressure and wind. Most of the time, the different data sources have both time and three spatial dimensions, and important interactions can happen on anything from a global scale to a micro perspective, and might follow different cycles. A method and resulting model that wants to achieve reasonably good results needs to understand all of these complex interactions. Many of these interactions are of a chaotic nature, making them very hard to predict [1].

Machine learning is currently one of the most common methods besides statistics to make sense of these data. In particular, deep learning has become one of the most popular methods, where part of the reason for this popularity is that deep learning allows you to learn from large amounts of data without the problems of over-fitting or suffering from high dimensional data, which often happen with classical machine learning methods. Further, deep learning is very good at extracting features from data. Each layer in a neural network can extract a different set of features, which eliminates the need for manual feature engineering. These layers are also able to perceive features that are not apparent to the human eye. Finally, more complex non-linear functions can usually be modelled by making the deep neural network deeper. One big disadvantage with the current development in using deep learning is that most methods are complex, and often unnecessary modifications of base architectures that overfit on the specific dataset at hand. All these specific attributes make deep learning specifically interesting for applications in connection to remote sensing. For example, vessel detection from satellite images, weather forecasting or climate change predictions.

To be able to handle the different types of data needed to solve the specific problem of weather prediction, multimodal machine learning (looking at different data streams with different properties at the same time) is needed. Multimodal deep learning is not very common, although it has gained attraction recently based on the success of models such as DALL-E that combine different modalities to perform specific tasks better than using only a single modality. However, in the context of weather prediction, most of the proposed work still focuses on single modalities.

Numerical Weather Prediction (NWP) models are commonly used in operational weather forecasting nowadays. These are complex mathematical models that seek to mimic the future conditions of the atmosphere on a three-dimensional grid using fundamental rules of ordinary physics. For processes that are smaller than the resolution of the model grid, the NWP models make several approximations. However, technological advancements have made it possible to run these models with higher spatial resolution, resulting in more detailed projections.

One of the essential facets of weather prediction is precipitation forecasting, which has also been the focus of much of the work within deep learning aimed at weather forecasting. A pivotal paper from this genre is [2] who proposed the convolutional LSTM for precipitation prediction for nowcasting—forecasts on a very short time horizon. Other examples include [3] who applied multi-task convolutional networks to the same problem, and [4,5] who developed MetNet and its successor MetNet-2, respectively—highly complex deep neural networks, which performed better than the physics-based models currently operating in the United States. A body of work is also dedicated to precipitation forecasting using UNets. Examples include [6] who used a UNET convolutional network to perform high-resolution precipitation forecasting from radar images, ref. [7] who developed the Temporal Recurrent U-Net and [8] who introduced SmaAt-UNet, a UNet-based convolutional neural network equipped with attention modules and depthwise-separable convolutions.

Temperature forecasting has also been receiving increasing attention from the deep learning community in recent years, with a wide range of approaches being explored. For example, ref. [9] applied a deep neural network with Stacked Denoising Auto-Encoders to historical temperature observations, ref. [10] explored the use of stacked LSTMs and [11] took the convolutional LSTM networks introduced by [2] and applied them in a temperature forecasting context. Ref. [12] performed a comparison between Stochastic Adversarial Video Prediction [13], Generative Adversarial Networks [14] and Variational Auto-Encoders [15]. An extensive review of neural networks used in temperature forecasting between 2005 and 2020 was performed by [16], with some of the more recent examples cited including [17] who compared a support vector machine, an artificial neural network and a time series based recurrent neural network, ref. [18] who compared a multi-layer perceptron, long short term memory network (LSTM) and a combination of convolutional neural network and LSTM and [19] who tackled this spatio-temporal problem using convolutional recurrent neural networks.

It is possible to imagine a variety of approaches to incorporating machine learning into weather forecasting or even the NWP models themselves. Most of the sources referenced above concern themselves with the post-processing of NWP model output. A few have sought to replace the NWP models entirely. The goal of this study is to approach the weather forecasting challenge as a multimodal data analysis problem by employing a new, lightweight deep learning method to improve short-term temperature forecasts using NWP model output merged with historical observations. The advantage with the new architecture is that it is designed as a base architecture similar to LSTM, etc, and that it can be expanded in the future with more complex designs. In deep learning, multimodal data processing has not been thoroughly investigated. The majority of methods employ early or late fusion for analysis, which involves merging the characteristics before analyzing them, or analyzing them independently and then combining them. Both methods have their own set of advantages and disadvantages.

In [20] we started exploring the possibilities of a new core architecture for multi-modal data analysis with the use case of weather prediction. This article builds upon this preliminary work.

The problem of temperature forecasting is totally dominated by different complex variants of CNN and ConvLSTM [16]. There are some exceptions, such as, for example, Gong et al. ([12]) who suggested the stochastic adversarial video prediction model. However, adversarial methods and other more recent methods are far larger and far more challenging to train than the tower networks presented here. In addition, these methods are often very complex and overfitted on the specific dataset they are developed for and they are not realistic to smaller datasets or to datasets with rather short time horizons, like the dataset used in this paper (2014–2018). To tackle this, we propose a new core architecture that is light weight and is easier to train and compare to the other core architectures in the domain, which are CNN and ConvLSTMs.

Specifically, we have added more and extensively extended experiments, new algorithms for comparison and new visualizations of the predictions. Besides demonstrating the usefulness of the proposed method we also show that it is more robust than the other methods. In addition, it is important to point out that the proposed architectures can also be used for any other multi modal data problem.

## 2. Materials and Methods

### 2.1. Data

Two datasets are used in this paper. The first is a set of 5 years of hourly 2 m air temperature observations from a dense network of official and quality controlled citizen weather stations interpolated to a grid with 1 × 1 km spatial resolution. More information about this dataset can be found in [21]. We have five years of data, covering a period from 2014 to 2018. The full grid covers all of Norway, however, a smaller subset of 40 × 40 grid points centered around Oslo has been used throughout this work. This area, shown in Figure 1, contains both an urban environment, forest and part of the Oslofjord inlet, with altitudes ranging from 0 to 647 above sea level.

The second dataset comes from an NWP model. The data are originally on a grid with a spatial resolution of 2.5 × 2.5 km [22], but have been regridded to the same 1 × 1 km grid as the gridded observations. The NWP data cover the same time period as the observational data, and the temporal resolution is hourly, however, fresh forecasts are only available four times a day, at 00, 06, 12 and 18 UTC.

From both datasets, we have altitude and land area fraction. It is well known that temperature generally decreases with increased altitude, and that there are often considerable differences in air temperature between land and over open water. These topographical data might be used to bridge discrepancies between the two datasets. In the second part of the work, we use these additional data as well as the day of the year as auxiliary predictors in the models.

While interpolated to the same grid in order to facilitate joining, the two datasets are fundamentally quite different. The NWP data are a gridded simulation of future states of the atmosphere, based on mathematical equations solved in space and time. The observations are point data—measurements from weather stations spread out irregularly in space.

From February 2018, we also have access to the official temperature forecasts from yr.no, a weather forecasting website and app hosted by the Norwegian government-owned national broadcasting corporation (NRK) and the Norwegian Meteorological Institute. The yr.no forecasts are based on the aforementioned NWP model data, but have gone through various forms of post-processing, and thus represent the state-of-the-art when it comes to weather forecasting.

Data and code can be made available upon request.

### 2.2. Models

The object of this work is the prediction of temperature from 1 to 6 h into the future. The work is twofold. Firstly, we investigate the usefulness of what we call the tower network from a weather forecasting perspective. Secondly, we compare the tower network to similar and state-of-the-art machine learning approaches.

The primary focus of this paper is a new deep learning model architecture called the tower network. The basic concept of this type of neural network is that it is built up of so-called “towers”—stacks of convolutional layers, batch normalization layers, max pooling and activation layers (see Figure 2). In theory, each tower should learn a slightly different view on the input data. These different views are achieved by varying the stride and/or kernel size in the convolutional layers. Different stride lengths were included in the experiments, which would allow for more efficient computations or downsampling, and we wanted to explore how this would impact the prediction performance [23]. Output from the towers is sent to a concatenation layer and finally to a convolutional layer. The motivation for testing these networks on a weather forecasting problem is that the weather is determined by relationships on several different spatial and temporal scales which makes it a very good use case for testing the new architecture we propose. These relationships are something one might hope to capture through the different views learned by the towers.

A modified version of the tower network is also tested, in which NWP data are provided directly to the concatenation layer (see dashed lines in Figure 2). Since the observations are historical and the NWP data are predictions of the future, the combination of these data is not necessarily completely straightforward. Layers equivalent to another tower are therefore added between the concatenation layer and the final convolutional layer to give this modified model the chance to combine the data in a meaningful way.

The tower networks are compared against a first order autoregressive model, AR(1) which is a traditional statistical approach, in which the observation $X_{t}$ at time *t* can be expressed in terms of the observation in the previous time step in the following way:$$\begin{array}{c} X_{t}=C+\varphi X_{t-1}+\varepsilon_{t} \end{array}$$
where *C* is a constant, φ is a regression parameter and $\varepsilon_{t}$ is white noise. The AR(1) is trained and evaluated in each grid point separately.

In the second part of the work, the strengths and weaknesses of the tower network are explored in comparison with two other machine learning approaches. The first is the convolutional neural network (CNN). As one of the cornerstones of deep learning for image processing, the CNN is an obvious choice for this comparison. It is a relatively simple, but powerful approach, and might give an indication of whether or not a more complex network is really needed for the prediction problem at hand.

The convolutional LSTM (convLSTM) has long been among the top performing deep learning approaches used for video prediction, and is thus a good representative of state-of-the-art deep learning methods, as well as an example of methods previously explored within the context of weather forecasting [2,11].

## 3. Results

Since there is five years of data, 2014–2016 are used for training, 2017 is used for validation, and 2018 for testing. The validation data are used in the training phase, to monitor the improvement of the models and to choose the appropriate stopping time. Testing data are data never before seen by the models, on which the models are finally tested. It is the results of these tests that are used in the comparison of models. Each sample of input data consists of 6 + 6 h of temperature observations and 6 h of NWP temperature forecasts. Figure 3 shows the structure of a sample: the observations that are used are from the input times, while the NWP forecasts correspond to the output times. In order to avoid the issue of diurnal trends that occur in temperature data, the model predicts for the same 6 h of every day, namely from 13 UTC to 18 UTC. The NWP data used are from the 12 UTC production time.

The data from yr.no are available only from 19 February 2018, so for comparisons with yr.no, the period from 19 February 2018 to 31 December 2018 is used in the evaluation.

### 3.1. Comparison 1: Weather Forecasting Baselines

In this first part of the work, three versions of the tower network are evaluated: one is trained solely on observational data, and is referred to as the observation based tower network, the second is trained solely on NWP model data, and is referred to as the NWP based tower network, and the third uses both types of input data, and is referred to as the multimodal tower network. The main reason for these experiments is to get a better and deeper understanding of the tower networks themselves and how different parameters affect the performance.

The parameters used in the tower networks are shown in Table 1. The training was done with Adam (adaptive moment estimation) as the optimizer, mean squared error as the loss function, a batch size of ten, and the number of epochs chosen using early stopping.

The tower networks are compared to persistence, which is a common baseline in short term weather forecasting. Persistence is the idea that the weather does not change, and this hypothesis often performs quite well on very short time horizons. In this work, the last observation in each grid point has been used as a forecast for the following 6 h.

Furthermore, the tower networks are compared to the raw NWP forecasts (the forecasts used as input to the networks), and the data from yr.no, which are the best available post-processed versions of the same data.

Lastly, a first-order autoregressive model has been trained on the temperature observations, in order to show whether the use of a neural network adds value beyond what can be achieved with a simple yet effective statistical method.

Figure 4 shows the root mean squared error (RMSE) of each model, averaged over times and locations, for each forecast hour. For the first two hours, persistence and the autoregressive model, AR(1), have the best performance, but their error rapidly grows and they perform the worst at the end of the 6 h period. This behaviour is unsurprising, since both models rely heavily on the last observation. The observation based tower network shows signs of the same trend, beginning with a small RMSE, that grows with time. However, the RMSE is neither as small at the beginning of the period nor as large at the end as that of persistence or AR(1). These findings suggest that the use of a neural network extracts more information from the observations than a simple statistical approach. It is, however, interesting that the tower network is not able to achieve the same performance in the first two hours.

The RMSE of the NWP forecasts exhibits a very different behaviour. It starts out quite large and decreases over time, while its overall shape is relatively flat. The NWP model combines information about initial conditions with an approximation of the behaviour of the atmosphere, and can therefore provide reliable predictions on a much longer horizon than persistence. The NWP based tower network adds little to no value to the raw NWP, suggesting that there is not a great deal of untapped potential in the NWP forecasts themselves.

The multimodal tower network is overall the best performer in this comparison. It cannot compete with persistence for the first 2 h, but has a relatively small RMSE for the whole 6 h period. The performance of the multimodal tower network compared with the networks trained only on historical observations or NWP model data indicates that the combination of these two data types gives more valuable insight than using them separately.

Finally, yr.no is included in this comparison since these are the data that the average person will see when they check the weather forecast on their phone. The weather forecasts on yr.no are optimized with respect to other criteria than simply RMSE, and thus it would be an oversimplification to say that the multimodal tower network is better. However, the multimodal tower network performs better in this test, which is a strong indication of its quality and potential as a post-processing technique.

While scores like the RMSE can give a relatively good indication of the quality of weather forecasts, there is also a need for the forecasts to look and behave realistically in space. Figure 5 shows the ground truth and predictions for the coldest day in the test dataset. This day was chosen not only as an example of the predictions in general, but also as an instance of something other than the average case. Since the models are optimized with respect to mean squared error, an unwanted side effect might be poorer performance in the less common, more extreme cases. In Figure 5, the ground truth shows a day that starts out at temperatures between −6 and approximately −11 degrees, and then becomes cooler over the course of the forecast period. Persistence, AR(1) and the observation based tower network all predict the temperatures of hour 1 quite well, but fail to capture the decrease in temperatures. It is also striking that the observation based tower network has far less detail than persistence and AR(1). The NWP, on the other hand, generally has the correct temperature development, but it is easy to see that it has been interpolated from a much coarser grid, and thus lacks the finer details of the other models. While the NWP based tower network appears smoother than the raw NWP, many of the finer features are not well represented. Both the multimodal tower network and yr.no successfully forecast the temperature decrease, as well as pinpointing areas with particularly high or low temperatures better than the other models.

### 3.2. Comparison 2: Deep Learning Approaches

In the second part of the work, the tower network was compared to the relatively simple but powerful CNN, and state-of-the-art in video and also weather prediction: the convLSTM. In this continuation of the work, we also tested the use of input data related to topography and time of year. It is important to point out that we are not aiming to compare specific variations of basic architectures like [16,24,25,26] on different datasets but rather want to compare to the basic architectures themselves. In addition, these methods usually have a lot of parameters and will result in over-fitting on our dataset with a short time horizon.

The architecture of the CNN is shown in Figure 6 and the convLSTM is shown in Figure 7. Since the data have been interpolated (in the case of the observations) and regridded (in the case of the NWP model data) to the same grid, it is technically possible to combine them in different ways.

In the case of the CNN, we have simply stacked all of the different inputs—observations, NWP data and auxiliary information—into 40×40×24 tensors. For the convLSTM, we have taken advantage of the possibility of having channels, and treated observations from the day prior to prediction, observations from the day of prediction and NWP data as three separate channels. A sample of these data has the shape 6×40×40×3. The auxiliary data were not used in this model because they did not appear to improve the performance of the convLSTM.

With the introduction of the auxiliary data, and the new data structures for the CNN and convLSTM, we also decided to re-examine the tower network architecture. We found that for the tower network, stacking observations and NWP data, like in the case of the CNN, and keeping the auxiliary data separate, produced the best predictions. See the new architecture in Figure 8, with corresponding parameter values in Table 2. The parameters of Tower 1* are the same as the parameters of Tower 1.

The three approaches were compared in terms of RMSE, training time and memory usage. In order to get a robust comparison, 15 realizations of each model were trained, i.e., 15 CNNs, 15 convLSTMs, and 15 tower networks. For each realization, training time, maximum memory usage and RMSE on the test data were recorded.

Figure 9 shows a box plot of the RMSE of the 15 realizations of each model, according to forecast hour. What is most striking about this figure is how the error of the CNNs and tower networks start out small and become larger, while the error of the convLSTMs behaves very differently, starting out quite large and ending up approximately the same as the other models, with no statistically significant difference between models for the last 3 h. The CNNs have the overall smallest errors, and among the tower networks, there is an outlier whose performance is consistently far worse than any other model. In general, however, the CNNs and the tower networks perform similarly to one another, and the difference between them is not statistically significant. The variability between the convLSTMs is quite small, relative to the other two models. This small variability, in combination with the generally larger errors for the first few hours, suggests that something slightly different is learned by these networks.

Figure 10 shows the training time, in minutes, per epoch for each model. The convLSTMs stand out as considerably slower than the two other models, which is also reflected in the total training times, an overview of which can be found in Table 3. The CNN is much quicker than the other models, using on average less than a minute per epoch. The tower networks fall somewhere in between, much faster than convLSTM but not as quick as CNN.

When it comes to maximum memory utilized in training the models, shown as a violin plot in Figure 11, there is considerable variability for all the models. However, the CNNs consistently require more memory than the two other models, and the convLSTMs require less than the other two. Here, it is worth mentioning that the CNNs and tower networks were trained with altitude, land area fraction and day of the year as auxiliary inputs, while the convLSTMs were not. This will have affected the memory usage.

Finally, we have looked at the validation error curves of the different models. These curves are shown in Figure 12 as moving averages (n=5), where we first averaged over all realizations. Since early stopping has been used, the number of realizations is not the same for all epochs, starting at 15 and declining as more realizations reach their final epoch. In this figure, we see that the validation error decreases at first, before leveling out, for all three models. This decrease happens more quickly for the convLSTMs than for the tower networks. However, while the tower networks keep slowly improving, the convLSTMs generally stop improving and we begin to see indications of over-fitting after a few hundred epochs. The tower networks also appear to start over-fitting towards the end. When it comes to the CNNs, their validation error initially decreases quickly, before almost plateauing around the same value as that of the convLSTM, slowly edging their way downwards.

In Figure 13, the best realization of each model (chosen on the basis of validation error) is compared alongside the meteorological baselines from the first part of the work. All three models outperform the raw NWP for every hour, as well as persistence from approximately hour 3. The line showing the error of the convLSTM is conspicuously parallel with the line for the raw NWP, while the CNN and tower network are both on par with, or outperforming, yr.no for every hour, approaching, but still nowhere near, persistence in hour 1.

We found it interesting that the convLSTM did not perform as well as the other models, and a quick look at some of its predictions suggested that it might struggle with the finer details—its resolution occasionally resembling that of the raw NWP. This issue was further investigated through permutation feature importance measurements, a concept introduced by [27] and generalized by [28].

Figure 14, Figure 15 and Figure 16 show feature importance as a function of hour for the three models. We immediately see that same day observations are a far more important feature for the CNN and the tower network than for the convLSTM. These findings support the theory that the convLSTM relies heavily on the NWP data, while not making full use of the historical observations. The CNN appears to place more value on previous day observations than the tower network does. The CNN is also the network whose performance is the most affected by feature perturbation.

## 4. Discussion

As exemplified in the previous section, the tower network performed well compared to meteorological baselines and simple statistical approaches. In addition, we showed that it also can complete with the current state of the art core architectures for the use case of weather forecasting. The comparison with other deep learning core architectures revealed a more nuanced picture, with the advantages of the tower network being less apparent. For example, CNN was the indisputable winner of the comparison of training times, while the convLSTM model required the least memory. With regards to both of these metrics, the tower network performed better than one model, but worse than the other. From a pure resource standpoint, there are no clear conclusions to be drawn, and depending on the user’s requirements, any one method might be preferable to the others. However, training time and memory usage are metrics of a secondary nature.

RMSE is what informs us of the forecasting performance of the models. In this comparison, we saw that the performance of convLSTM in the first 3 h was considerably worse than the CNN and tower networks, suggesting a disproportionate reliance on the NWP data. This suspicion was further supported by the permutation feature importance measurements, which also revealed that the convLSTM, unlike the other two models considered, valued previous day observations over same day observations. There was no significant difference in RMSE between the CNNs and tower networks. However, studying the permutation feature importance measurements for all 15 realizations of each of these models revealed that permuting any one input data source had an overall greater impact on the RMSE of the CNNs than on that of the tower networks, which might suggest a greater robustness in the tower networks. This makes the tower networks a useful addition to the other benchmarked core architectures and suggest that the tower networks are more robust and generalizable. For future work, it would be interesting to explore how the tower networks can replace the other two core architectures in more complex networks, and how this influences the generalizability and robustness of these. In addition it would also be interesting to test the tower networks with a variety of other data types and for other use cases.

## 5. Conclusions

In this paper, we have seen an example of how deep learning can be useful in the post-processing of weather forecasts, and produces results that are comparable to, and often better than, traditional methods. Furthermore, we have compared three approaches, the CNN, the convLSTM and the tower network. Each method had strengths and weaknesses: the CNN was quick and performed excellently, but was also memory intensive; the convLSTM was slow to train, and did not perform quite as well, although it required the least memory; and the tower network was quick, well performing and lightweight, but not significantly better than the simpler, and quicker, CNN.

The first conclusion of this paper is that the proposed new core architecture is more robust and generalizable than its currently used counterparts. The second conclusion is that when using deep learning in weather forecasting, good results can be achieved with computationally inexpensive methods, which suggests that most of the complex variations in core architectures are most probably over-fitting or wasting unnecessary resources. The third conclusion is that combining NWP data and historical observations as input to the neural networks creates better results than using only one of these data types alone. We have seen that with deep learning, we can effectively combine various data sources and, crucially, we found that the benefit of combining the data was far greater than the advantage of one method over another. In fact, all the methods considered in this paper perform reasonably well when given this combined input.

For the new tower network introduced in this paper, the greatest advantage appears to be its ability to extract the essential information from heterogeneous inputs, which would be interesting to explore in future work and other domains.

## Figures and Tables

**Figure 1 sensors-22-02802-f001:**
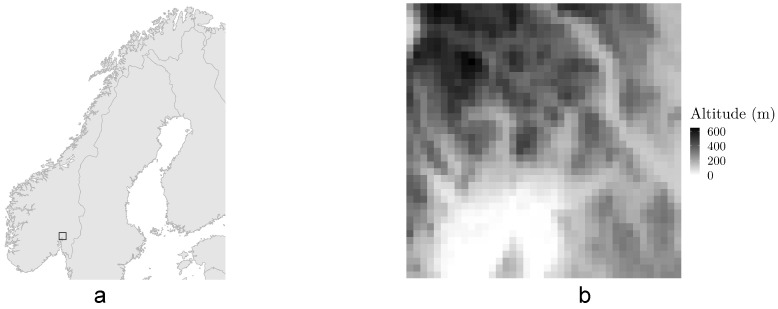
(**a**) The geographical area used in this work, shown as an empty, black square centered around Oslo on a map of Scandinavia. (**b**) The topography of the area used.The Oslo fjord inlet can be seen towards the bottom of the image.

**Figure 2 sensors-22-02802-f002:**
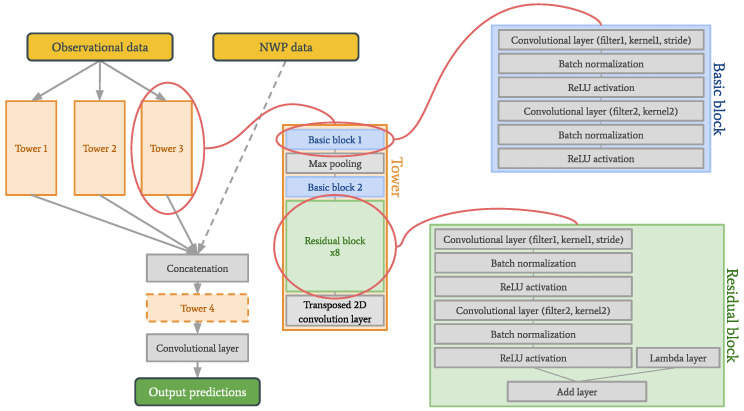
Network architecture for the normal and the modified tower network. The observational data have the dimensions 40×40×12, and the NWP data 40×40×6. The specific parameters used in this work are listed in Table 1.

**Figure 3 sensors-22-02802-f003:**
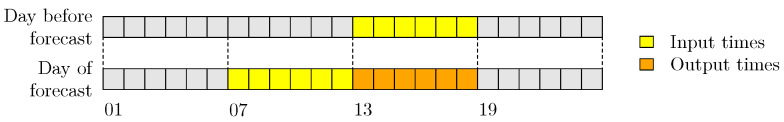
One sample of observation data. Historical observations are observations from the input times, shown here in yellow. The NWP data are forecast data valid in the output times, here shown in orange.

**Figure 4 sensors-22-02802-f004:**
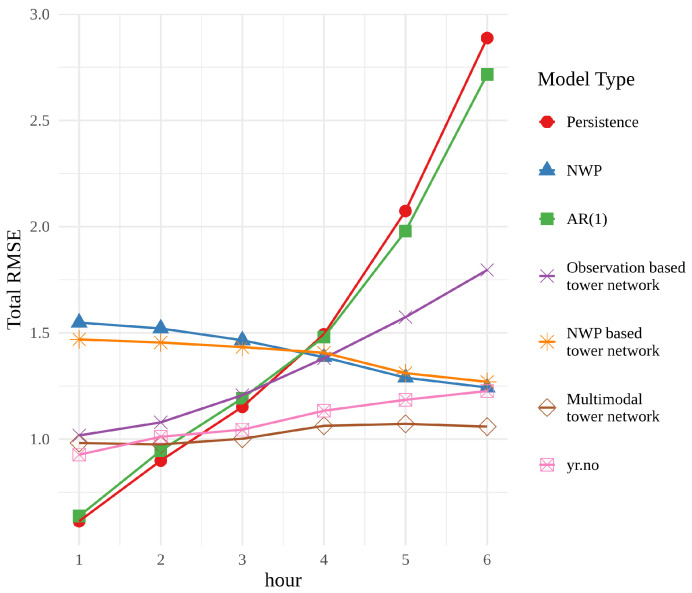
Root mean squared error of the tested models and meteorological baselines averaged over the spatial grid.

**Figure 5 sensors-22-02802-f005:**
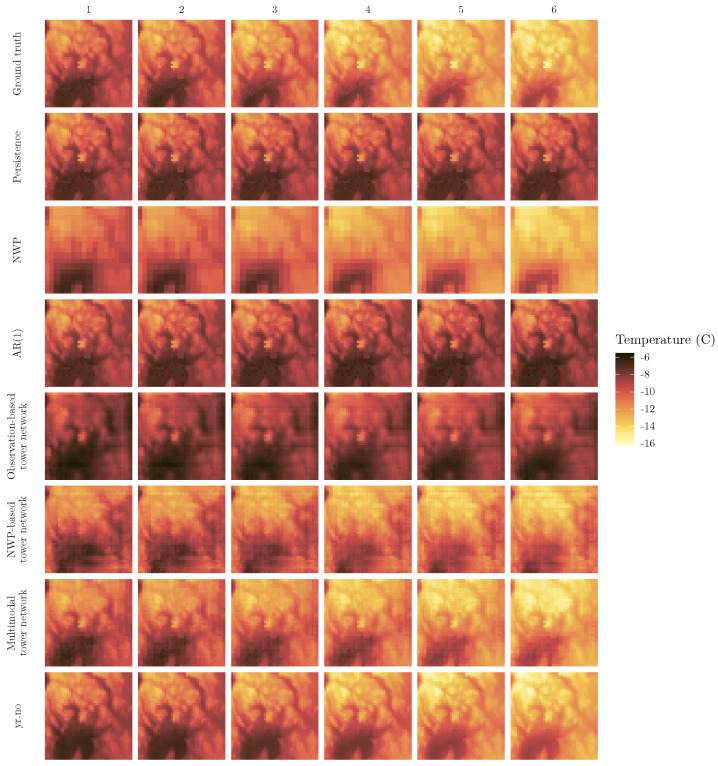
Example of temperature forecasts from the different models with the ground truth for reference. Each row corresponds to a model, and each column to an hour.

**Figure 6 sensors-22-02802-f006:**
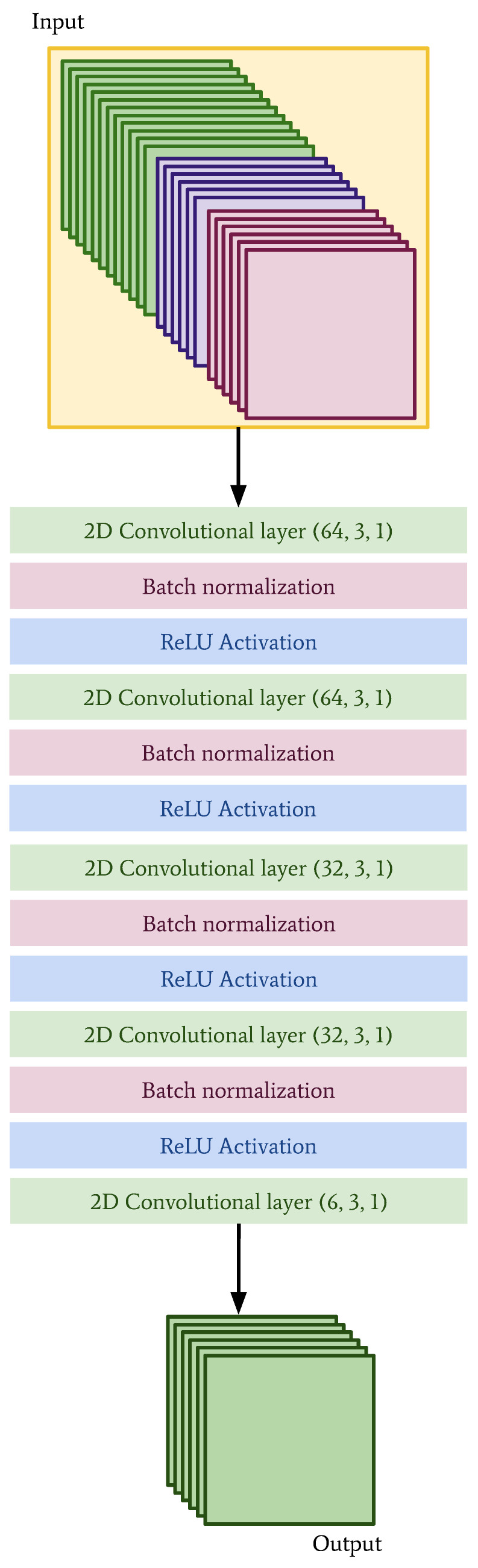
Network architecture for the CNN. The input data are made up of historical observations, NWP forecasts and auxiliary inputs such as land area fraction and altitude. The inputs are stacked, with the resulting dimensions being 40×40×24.

**Figure 7 sensors-22-02802-f007:**
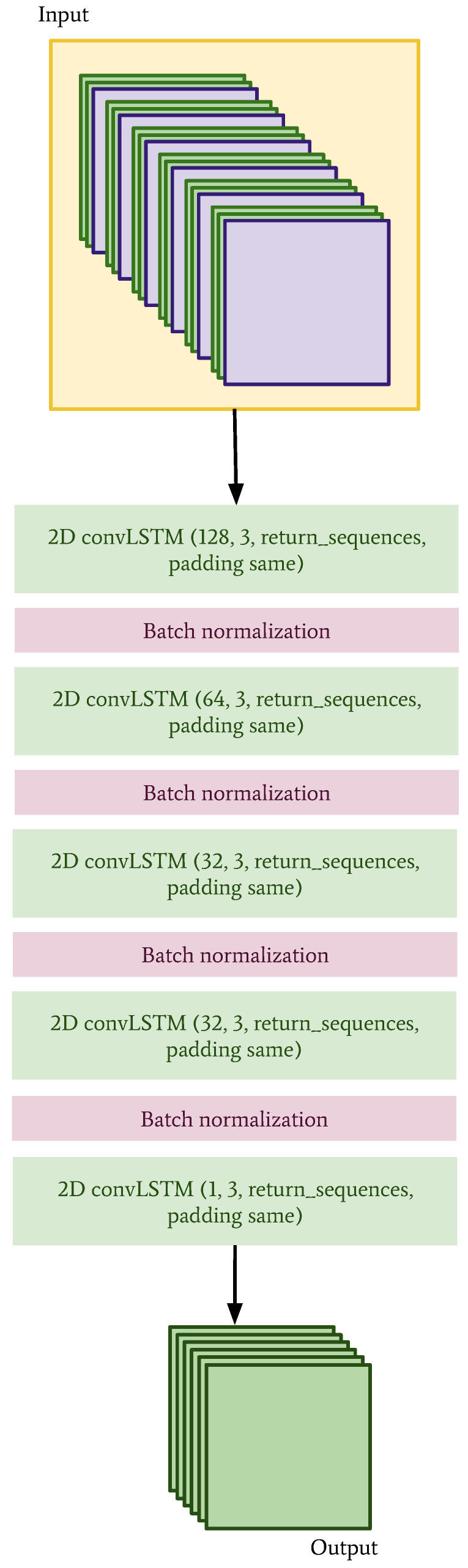
Network architecture for the convolutional LSTM. The input data are made up of historical observations from the previous day, historical observations from the 6 h prior to the predicted times, and NWP forecasts. These three inputs are treated like channels, such that the resulting dimensions of the input data are 6×40×40×3.

**Figure 8 sensors-22-02802-f008:**
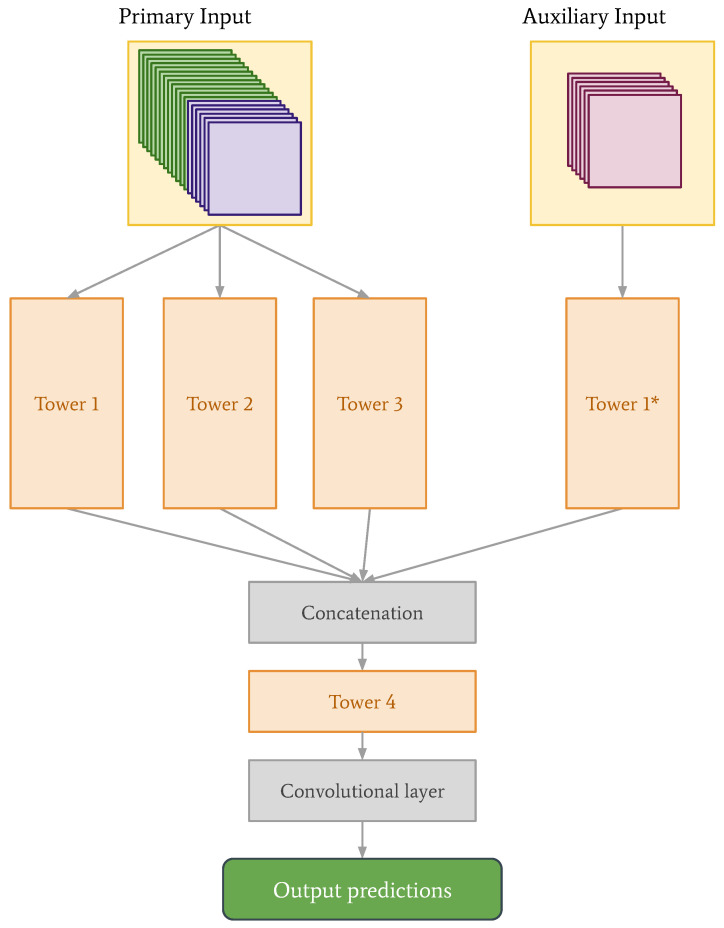
Network architecture for the tower network with primary input made up of stacked observational and NWP data (40×40×18), and auxiliary input made up of land area fraction and altitude from the observations and NWP data set, as well as sine and cosine values corresponding to day of the year mapped to values between 0 and 2π (40×40×6). The construction of a tower remains the same as what is shown in Figure 2.

**Figure 9 sensors-22-02802-f009:**
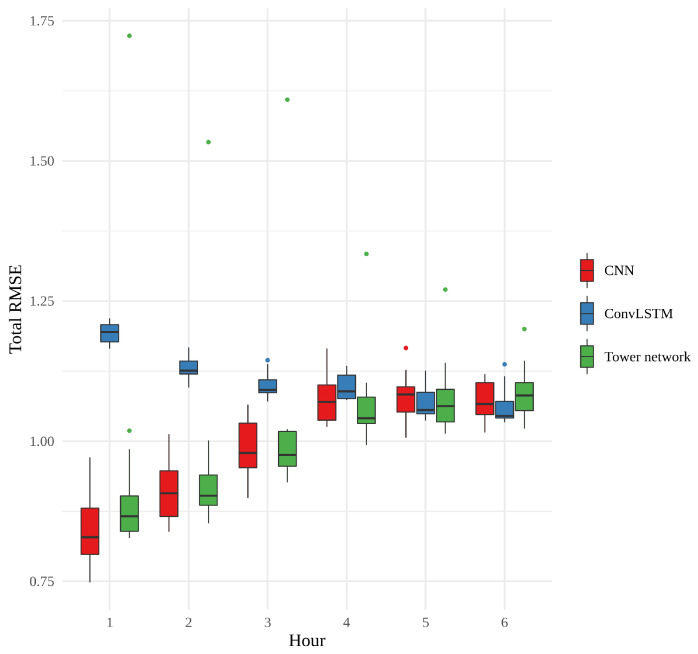
Boxplot showing the root mean squared error of the 15 realizations of each model, averaged over the spatial grid.

**Figure 10 sensors-22-02802-f010:**
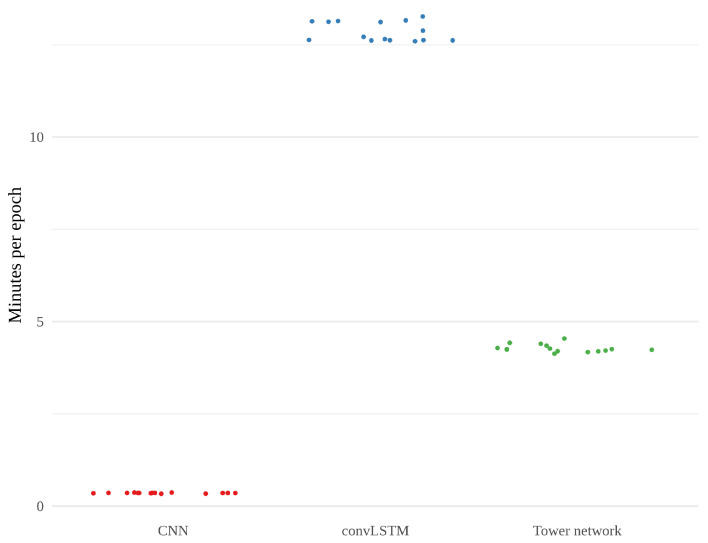
Training time per epoch for the 15 realizations of CNN, convLSTM and tower network in minutes.

**Figure 11 sensors-22-02802-f011:**
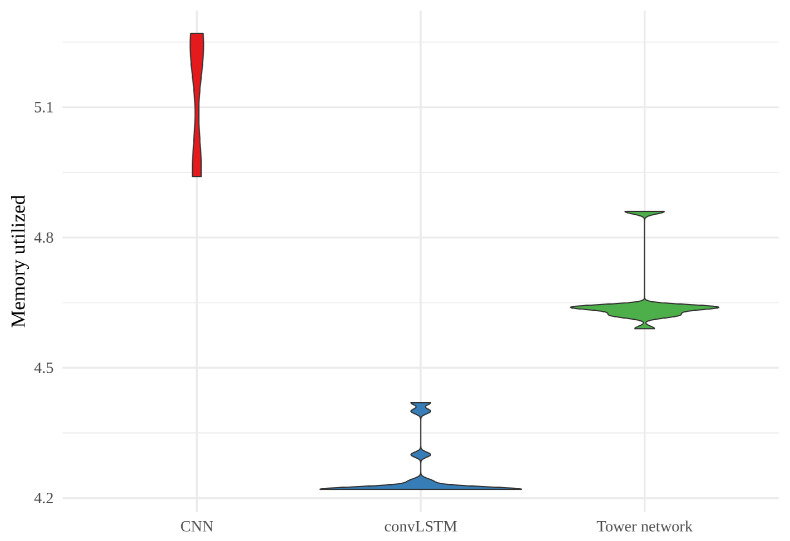
Violin plot of the memory (in GB) utilized when training the models. Each violin represents the memory usage of 15 realizations of a model. From the left, there is CNN in red, convLSTM in blue and the tower network in green.

**Figure 12 sensors-22-02802-f012:**
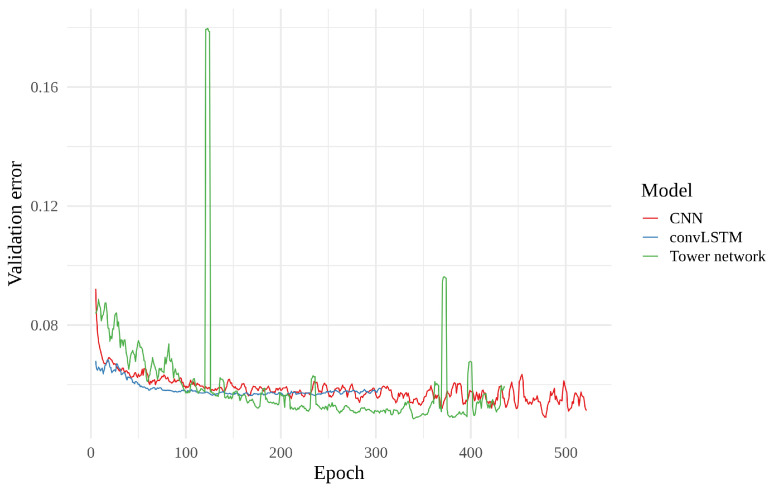
Moving average with a window of 5 of the normalized validation error of each model as a function of number of epochs. Values are averages for the 15 realizations of each model type.

**Figure 13 sensors-22-02802-f013:**
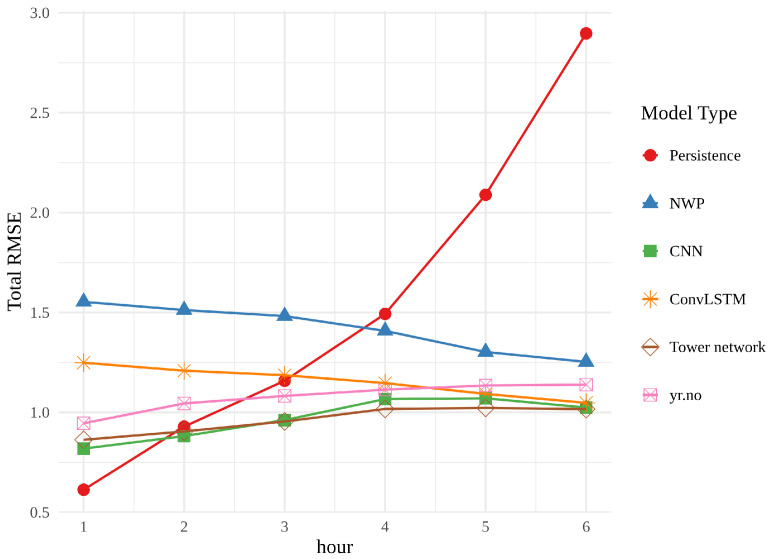
Comparison of the RMSE of the best realization of each neural network and meteorological baselines, all averaged over the spatial grid.

**Figure 14 sensors-22-02802-f014:**
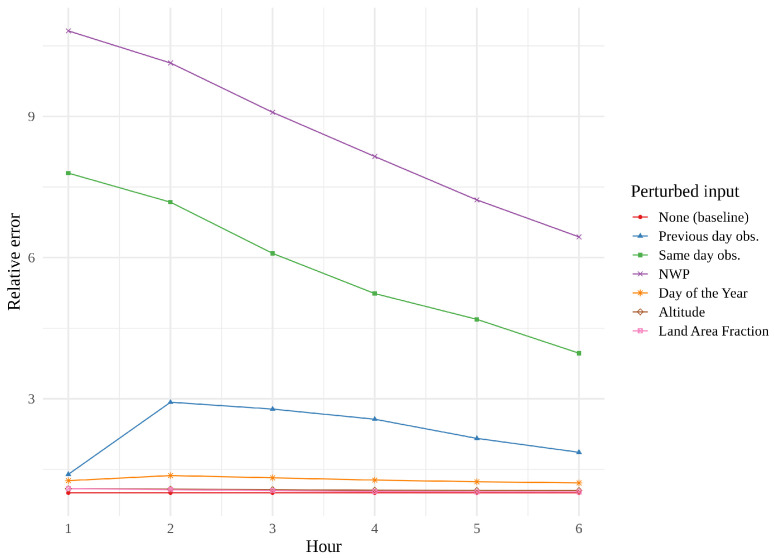
Permutation feature importance of the best CNN, i.e., the error resulting from the permutation of one parameter, leaving the remaining parameters untouched.

**Figure 15 sensors-22-02802-f015:**
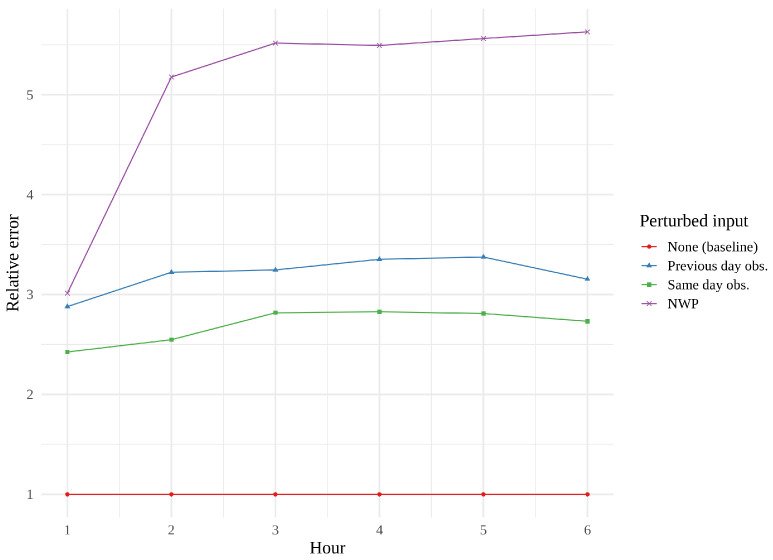
Permutation feature importance of the best convLSTM, i.e., the error resulting from the permutation of one parameter, leaving the remaining parameters untouched.

**Figure 16 sensors-22-02802-f016:**
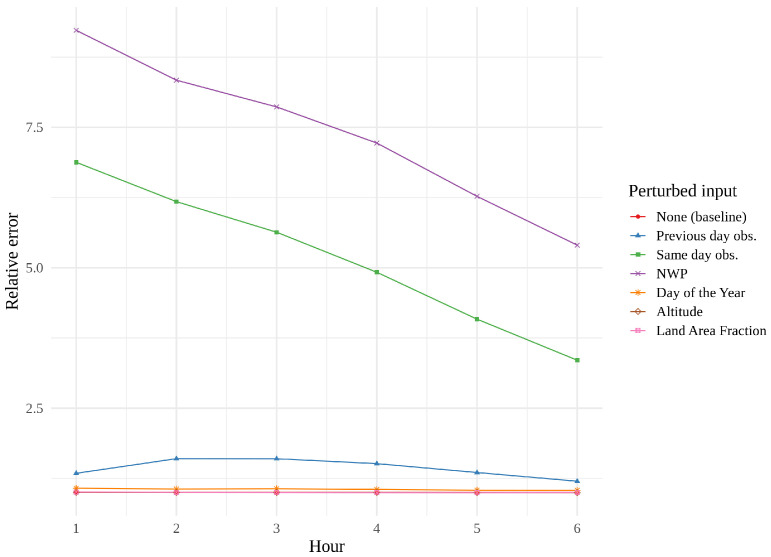
Permutation feature importance of the best tower network, i.e., the error resulting from the permutation of one parameter, leaving the remaining parameters untouched.

**Table 1 sensors-22-02802-t001:** All network parameters for the tower networks in the first part of the work.

		Filters 1	Filters 2	Kernel Size 1	Kernel Size 2	Strides
Tower 1	Basic block 1	64	32	8	3	1, 1
	Basic block 2	64	32	3	3	1, 1
	Residual blocks	64	32	3	3	1, 1
	Transposed 2D	64	-	4	-	2, 2
Tower 2	Basic block 1	64	32	8	3	1, 2
	Basic block 2	64	32	3	3	1, 2
	Residual blocks	64	32	3	3	1, 1
	Transposed 2D	64	-	4	-	2, 8
Tower 3	Basic block 1	64	32	8	3	1, 4
	Basic block 2	64	32	3	3	1, 4
	Residual blocks	64	32	3	3	1, 1
	Transposed 2D	64	-	4	-	2, 20
Tower 4	Basic block 1	64	32	8	3	1, 1
	Basic block 2	64	32	3	3	1, 1
	Residual blocks	64	32	3	3	1, 1
	Transposed 2D	64	-	4	-	2, 2
Final convolutional layer		6	-	8	-	-

**Table 2 sensors-22-02802-t002:** All network parameters for the tower networks in the second part of the work.

		Filters 1	Filters 2	Kernel Size 1	Kernel Size 2	Stride
Tower 1	Basic block 1	64	32	8	3	1
	Basic block 2	64	32	3	3	1
	Residual blocks	64	32	3	3	1
	Transposed 2D	64	-	4	-	2
Tower 2	Basic block 1	64	32	8	3	2
	Basic block 2	64	32	3	3	2
	Residual blocks	64	32	3	3	1
	Transposed 2D	64	-	4	-	8
Tower 3	Basic block 1	64	32	8	3	4
	Basic block 2	64	32	3	3	4
	Residual blocks	64	32	3	3	1
	Transposed 2D	64	-	4	-	20
Tower 4	Basic block 1	64	32	8	3	1
	Basic block 2	64	32	3	3	1
	Residual blocks	64	32	3	3	1
	Transposed 2D	64	-	4	-	2
Final convolutional layer		6	-	8	-	-

**Table 3 sensors-22-02802-t003:** Total training time in minutes.

	Min	Median	Max
CNN	1.01	1.64	2.91
convLSTM	34.8	45.7	64.9
Tower network	16.3	21.9	32.0

## Data Availability

All data and code is available by request from the corresponding authors.

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
