# Peer review of "Deep Tower Networks for Efficient Temperature Forecasting from Multiple Data Sources"

_sensors, 2022, doi:10.3390/s22072802_

Round 1
Reviewer 1 Report
The goal of this study is to approach the challenge as a multimodal data analysis problem by employing a new, lightweight deep learning method to improve short-term temperature forecasts using NWP models by merging them with historical observations.This paper investigates the usefulness of a novel method called tower networks. This method is able to learn from multiple different input data sources at once. The results of this paper indicates that the tower network performs well in comparison both with the meteorological baselines, and with the other machine learning approaches.
The paper is well written and well structured. However, I have the following concerns.
1- There are very strong existing methods to forecast the weather. The authors have to compare their proposed methods against state-of-the-arts (other methods tested in research papers) in this area. Only comparing the proposed method with CNN, LSTM, or different variants of the proposed methods would not suffice and prove that this approach is better than others.
2- It should be explained clearly, maybe by giving an example explaining the multi-modal data in this paper.
3- It should be explained clears what the motivation behind using tower networks in this weather forecasting problem.
Reviewer 2 Report
This manuscript is overall well designed for comparing several machine-learning methods used for the prediction of the weather-specific dataset. A minor to moderate revision is required to enhance the scientific soundness of this manuscript. My concerns are listed as follows,
- The abstract is somewhat short to highlight the innovatory design and new findings of this manuscript. At least, some key features such as the indices of accuracy assessment, training time, and memory usage. should be given. Besides, the potential applicability and limitation of this method should be concisely addressed.
- Figure 1 should incorporate the elevation table.
- Figure 7 needs to adjust its size since some annotations were missing.
- In Figure 9, whether there exist statistically significant differences between the three models during hours 4 and 6?
- Where is the discussion section? In this section, alongside the theoretical interpretation of the model performance, remarks on the innovatory design and new findings of this manuscript should be well addressed.
Reviewer 3 Report
In the Abstract, you mention that the tower network performs well compared to weather baselines and to the other machine learning approaches. Is this developed algorithm within the machine learning group? Expand the abstract, it is very synthesized
Pag. 2, line 74-76 you say: “There are a variety of approaches to incorporating machine learning into weather forecasting or even into the NWP models themselves” includes at least one reference.
Pag. 2, line 75-82: You must improve the writing of the objective of the work or, where appropriate, improve the translation. This statement is not clear “The objective of this study is to address the challenge as an analysis problem” … of multimodal data by employing a new and lightweight deep learning method to improve short-term temperature forecasts using NWP models by merging them with historical observations.
Page 3; line 93-94, You say: Two data sets are used in this document. The first is a 5-year set of hourly air temperature observations at 2 m interpolated on a grid with a spatial resolution of 1 x 1 km. Mention if it is the NOAA-AVHRR sensor, and why Landsat-8 was not used, since it has better spatial resolution. Explain.
The 5 years of data is a poor historical series, you should include a broader series, for example 2010-2021. Mention why a broader historical series could not be obtained. Include an explanation.
Page 3, line 104-105: How altitude and fraction of land area affect topographic discrepancies between the two data sets, mention it.
Page 5, line 148-149: If there were 5 years of data (2014-2018), 2014 - 2016 are used for algorithm training, 2017 is used for validation and 2018 for testing. What is the difference between validation and testing? Validation is testing with the support of the performance indicators achieved by the algorithm. It is necessary to improve the wording of this argument.
Page. 8, line 225-228: In the second part of the work, it is recommended to compare your algorithm (tower network) that belongs to the machine learning group with a Deep learning algorithm (CNN), if so, explain it and include a reference.
Page16 and 17, line 304-321: improve the writing of the conclusions, it is very confusing.
The first conclusion is not really a conclusion, it seems like a recommendation. With the Google platform, you can access the cloud to perform jobs that require speed and computational power.
The second conclusion is very ambiguous, you should improve the wording.
Finally you say that all the methods considered in this article
work reasonably well, it is necessary to show quantitative performance indicators that demonstrate the level of effectiveness of the tower network algorithm.
In general, it is recommended to improve the English translation and the writing of the entire manuscript.

Round 2
Reviewer 1 Report
This paper lacks enough novelty. It's only an implementation of a deep learning approach. Comparison with state of the art approaches is missing.
- In the first part of the paper the authors only compared their method with other tower networks. In the second part, they compared their method with CNN and ConvLSTM.
- There are lots of state of arts methods (published papers) within the scope of this paper. To show the proposed method actually works better than other methods, authors have to compare their method with other published papers.
Author Response
Sensors
Manuscript ID: sensors-1619048
Manuscript Title: Deep Tower Networks for Efficient Temperature Forecasting from Multiple Data Sources
Authors: Siri S. Eide, Michael A. Riegler, Hugo L. Hammer and John Bjørnar Bremnes
Dear editors and reviewers
Thank you very much for your valuable feedback. We have considered your suggestions and criticism and revised the paper accordingly. Please find our comments below. The new changes are marked with green in the revised document.
This paper lacks enough novelty. It's only an implementation of a deep learning approach.
Thank you for the comment. We think that our approach is substantially different from existing methods. In addition it will allow other researchers to build upon by improving the suggested method or applying it to other domains and use cases.
Comparison with state of the art approaches is missing.
As state of the art for your approach we assume the core architectures and not variations of them. This comparison we have performed. We added some text to the paper to make this point more clear.
- In the first part of the paper the authors only compared their method with other tower networks. In the second part, they compared their method with CNN and ConvLSTM.
The first experiment was designed to get a better understanding of the tower networks and explore the implications of different hyperparameters. We added a sentence to clarify this.
- There are lots of state of arts methods (published papers) within the scope of this paper. To show the proposed method actually works better than other methods, authors have to compare their method with other published papers
As mentioned above we do not want to compare these methods since we are only interested in the basic architectures. For sure it would be possible to try different hyperparameters and benchmark with other methods but we do not see what this would add to the message of the article. We added a paragraph to make our intention more clear and also referenced some work that adopted core architectures such as CNNs and convLSTMs.

This manuscript is a resubmission of an earlier submission. The following is a list of the peer review reports and author responses from that submission.